# The DiffuseStyleGesture+ entry to the GENEA Challenge 2023

Sicheng Yang*
Haiwei Xue*
Shenzhen International Graduate School, Tsinghua
University, Shenzhen, China
yangsc21@mails.tsinghua.edu.cn
xhw22@mails.tsinghua.edu.cn

Minglei Li[†]
Zonghong Dai
Huawei Cloud Computing Technologies Co., Ltd
Shenzhen, China
liminglei29@huawei.com
daizonghong@huawei.com

Zhiyong Wu[†]
Shenzhen International Graduate School, Tsinghua
University, Shenzhen, China
The Chinese University of Hong Kong
Hong Kong SAR, China
zywu@sz.tsinghua.edu.cn

Zhensong Zhang
Songcen Xu
Xiaofei Wu
Huawei Noah's Ark Lab, Shenzhen, China
zhangzhensong@huawei.com
xusongcen@huawei.com
wuxiaofei2@huawei.com

## ABSTRACT

In this paper, we introduce the DiffuseStyleGesture+, our solution for the Generation and Evaluation of Non-verbal Behavior for Embodied Agents (GENEA) Challenge 2023, which aims to foster the development of realistic, automated systems for generating conversational gestures. Participants are provided with a pre-processed dataset and their systems are evaluated through crowdsourced scoring. Our proposed model, DiffuseStyleGesture+, leverages a diffusion model to generate gestures automatically. It incorporates a variety of modalities, including audio, text, speaker ID, and seed gestures. These diverse modalities are mapped to a hidden space and processed by a modified diffusion model to produce the corresponding gesture for a given speech input. Upon evaluation, the DiffuseStyleGesture+ demonstrated performance on par with the top-tier models in the challenge, showing no significant differences with those models in human-likeness, appropriateness for the interlocutor, and achieving competitive performance with the best model on appropriateness for agent speech. This indicates that our model is competitive and effective in generating realistic and appropriate gestures for given speech. The code, pre-trained models, and demos are available at this URL.

## CCS CONCEPTS

• **Human-centered computing** → **Human computer interaction (HCI)**; • **Computing methodologies** → **Motion processing**; **Neural networks**.

---

*Both authors contributed equally to this research.
[†]Corresponding author

## KEYWORDS

gesture generation, diffusion-based model, conversation gesture

**ACM Reference Format:**
Sicheng Yang, Haiwei Xue, Zhiyong Wu, Minglei Li, Zonghong Dai, Zhensong Zhang, Songcen Xu, and Xiaofei Wu. 2023. The DiffuseStyleGesture+ entry to the GENEA Challenge 2023. In *Proceedings of ACM International Conference on Multimodal Interaction (ICMI '23)*. ACM, New York, NY, USA, 7 pages. https://doi.org/XXXXXXX.XXXXXXX

## 1 INTRODUCTION

Non-verbal behaviors, particularly gestures, act a crucial role in our communication [24]. They provide the necessary spark to animate robotic interfaces, encapsulate diverse functional information, and subtly deliver social cues. We can create more engaging, informative, and socially adept robotic systems by incorporating these behaviors. And gestures enrich communication with non-verbal nuances [24, 39]. Indeed, natural conversations often incorporate body gestures, which can lead to perceptions of dullness or unnaturalness if absent. Individuals use gestures to express ideas and feelings, either directly or indirectly. For instance, the formation of a circle using the thumb and forefinger—an open palm gesture—communicates the concept of "OK" [32].

3D gesture generation has drawn much attention in the community. Early studies leveraged unimodal inputs, Dai et al. [10] employ audio features to drive gesture synthesis via Bi-LSTMs, and some works incorporate GANs and VAEs to learn relevant pairs and improve synthesis quality [19, 26, 34]. However, these methods encountered challenges such as gesture diversity and training difficulties. On the other hand, some works also explored textual modality, Chiu et al. [6] introducing the DCNF model combining speech, textual content, and prosody, and Yoon et al. [38] proposing an Encoder-Decoder framework. Liang et al. [20] introduces SEmantic Energized Generation (SEEG), a novel approach that excels at semantic-aware gesture generation. Recently, multimodal methods [1, 9, 35, 37] integrating both audio and text have gained attention, focusing on the semantic feature encoding and long sequence modeling of 3D human motion. Further, many works begin

to pay attention to the speaker's identity [21, 22], style [8, 33], emotion [25, 36], etc. Despite significant advances, gesture generation using a comprehensive multimodal approach remains challenging, mainly due to the inherent trade-off between quality and diversity [33].

Recently, diffusion models [11] have shown great potential for generating motions [7, 29, 41], achieving high-quality outputs while maintaining diversity. Hence, in this gesture generation challenge, we attempt to apply diffusion models to tackle the problem of multimodal gesture generation.

Inspired by [33], we find that the diffusion model-based approach for co-speech gesture generation surpasses other deep generative models of motion in terms of quality and alignment with speech, while allowing for the generation of stylized and diverse gestures. In this paper, we incorporate textual modality using the DiffuseStyleGesture framework and restructure the architecture. Furthermore, we also refined the representations of gesture and audio, in alignment with the challenge dataset. These enhancements allow the model to generate high-quality, speech-aligned, speaker-specific stylized, and diverse gestures with significant controllability. We submitted our system to the GENEA challenge 2023 [16], which aims to consolidate and compare various methods for co-speech gesture generation and evaluation, promoting the development of non-verbal behavior generation and its evaluation via a large-scale user study involving a common dataset and virtual agent.

The main contributions of our paper are: (1) We propose DiffuseStyleGesture+, a multimodal-driven gesture generation model with improved input network structure, input modality and feature representation, as well as the diffusion model with cross-local attention. (2) The evaluation of the GENEA Challenge demonstrates that our model is among the first tier at human-likeness, appropriateness for the interlocutor, and achieves competitive performance on appropriateness for agent speech. (3) The ablation study validates the effectiveness of our proposed denoising module. Besides, we discuss the stylization and diversity of the generated gestures, as well as further discussion of more technical details.

## 2 METHOD

Our method is based on DiffuseStyleGesture [33], a recent diffusion model-based speech-driven gesture generation approach. Besides seed gesture, audio and speaker ID, we also take text as an additional input modality. The overview of this work is shown in Figure 1.

### 2.1 Feature Extraction

We extract the features of the input modalities as follows:

- Gesture: We used 62 joints including the fingers, and each frame represents the motion features in terms of position, velocity, acceleration, rotation matrix, rotational angular velocity, and rotational angular acceleration of each joint. Although there are certain relations between positions, velocities, accelerations, etc., which can be transformed into each other, representing motion features with more motion data can lead to better performance [8, 40]. We denote the natural mocap gestures clip as $x_0 \in \mathbb{R}^{(N_{seed}+N) \times [62 \times (9+3) \times 3]}$. The first $N_{seed}$ frames of the gestures clip $x_0$ are used as the seed

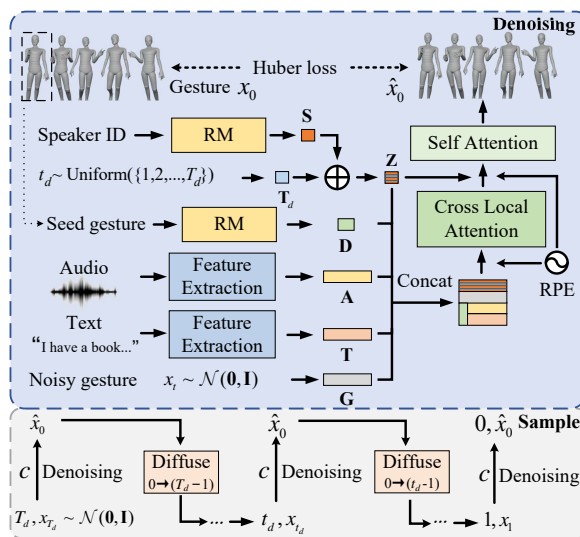

Figure 1: (Top) Denoising module. A noising step $t_d$ and a noisy gesture sequence $x_t$ at this noising step conditioning on $c$ (including seed gesture, audio, speaker ID and text) are fed into the model. (Bottom) Sample module. At each noising step $t_d$, we predict the $\hat{x}_0$ with the denoising process, then add the noise to the noising step $x_{t_d-1}$ with the diffuse process. This process is repeated from $t_d = T_d$ until $t_d = 0$.

gesture and the remaining $N$ frames are what the model needs to predict based on text and audio.

- Audio: More speech features also lead to better performance [4, 15]. Different representations can complement each other, e.g., representations such as pitch contain rhythmic content, the pre-trained model features such as WavLM [5] contain more complex information such as emotion, Onsets contain beat information, etc. We combine MFCC, Mel Spectrum, Pitch, Energy [39], WavLM [5], and Onsets [2] as audio features. We denote the features of audio clip as $\mathbf{A} \in \mathbb{R}^{N \times (40+64+2+2+1024+1)}$.

- Speaker ID: The ID of the speaker is represented as one-hot vectors where only one element of a selected ID is nonzero. The Talking With Hands dataset has a total of 17 speakers, so the dimension of speaker ID is 17.

- Text: Following [39], we use FastText [3] to obtain the 300-D word embeddings. And we use one bit to indicate whether there is a laugh or not, and the last bit is set to 0 as [4]. Each word is mapped to its pre-trained word embedding at word-level granularity. Then the features of text clip $\mathbf{T} \in \mathbb{R}^{N \times 302}$.

### 2.2 Gesture Denoising

Unlike text semantics-driven motion generation [13, 29, 41], they only need a *token* to contain the semantics of a sentence, which haven't to be aligned with time. Gesture generation is temporally perceptible, that is, the gestures are related to the rhythm of the speech. So we perform linear interpolation of the extracted audio

features $\mathbf{A}$ in the temporal dimension in order to align with the gestures. Gestures and music-driven dance generation [28, 30, 42] are also different. Gestures and semantics are also temporally related, for example, the hand opens when saying 'big'. As in [4, 37], we use frame-level aligned word vectors $\mathbf{T}$.

Our goal is to synthesize a high-quality and speech-matched human gesture $\hat{x}$ of length $N$ given conditions $c$ using the diffusion model [11]. Following [29], we predict the signal itself instead of predicting noise at each noising step $t_d$. As shown in the top of Figure 1, the Denoising module reconstructs the original gesture $x_0$ from the pure noise $x_t$, noising step $t_d$ and conditions $c$.

$$\hat{x}_0 = \text{Denoise}\left(x_{t_d}, t_d, c\right) \qquad (1)$$

where $c = [\mathbf{S}, \mathbf{D}, \mathbf{A}, \mathbf{T}]$. During training, noising step $t_d$ is sampled from a uniform distribution of $\{1, 2, \ldots, T_d\}$, with the position encoding [31]. $x_{t_d}$ is the noisy gesture with the same dimension as the real gesture $x_0$ obtained by sampling from the standard normal distribution $\mathcal{N}(0, \mathbf{I})$.

We add the information of the noising step $\mathbf{T}_d$ and speaker ID $\mathbf{S}$ to form $\mathbf{Z}$ and replicate and stack them into a sequence feature of length $N_{seed} + N$. The overall attention mechanism is similar to [33], using cross-local attention [27], self-attention [31] and relative position encoding (RPE) [14]. The difference is that we condition $\mathbf{D}$ in the first $N_{seed}$ frames and $\mathbf{A}$ and $\mathbf{T}$ in the last $N$ frames, so that the smooth transition between segments is considered in the first $N_{seed}$ frames and the corresponding gestures are generated in the last $N$ frames based on audio and text, which reduce the redundancy of inputs.

Then the Denoising module is trained by optimizing the Huber loss [12] between the generated gestures $\hat{x}_0$ and the real human gestures $x_0$:

$$\mathcal{L} = E_{x_0 \sim q(x_0|c), t_d \sim [1, T_d]} \left[\text{HuberLoss}(x_0 - \hat{x}_0)\right] \qquad (2)$$

### 2.3 Gesture Sampling

As shown in the bottom of Figure 1, when sampling, the initial noisy gesture $x_T$ is sampled from the standard normal distribution and the other $x_{t_d}, t_d < T_d$ is the result of the previous noising step. The final gesture is given by splicing a number of clips of length $N$. The seed gesture for the first clip is a gesture from the dataset. Then the seed gesture for other clips is the last $N_{seed}$ frames of the gesture generated in the previous clip. For every clip, in every noising step $t_d$, we predict the clean gesture $\hat{x}_0$ using Equation (1) and add Gaussian noise to the noising step $x_{t_d-1}$ with the diffuse process [11]. This process is repeated from $t_d = T_d$ until $x_0$ is reached.

## 3 EXPERIMENT

### 3.1 Experiment Setting

We trained on all the data in the GENEA Challenge 2023 [16] training dataset, which is based on Talking With Hands [18]. In this work, gesture data are cropped to a length of 150 frames (5 seconds, 30 fps), with the first $N_{seed} = 30$ frames as seed gesture, and the last $N = 120$ frames to calculate the loss between generated gestures and real gestures in Equation (2). We use standard normalization (zero mean and unit variant) to all joint feature dimensions. The

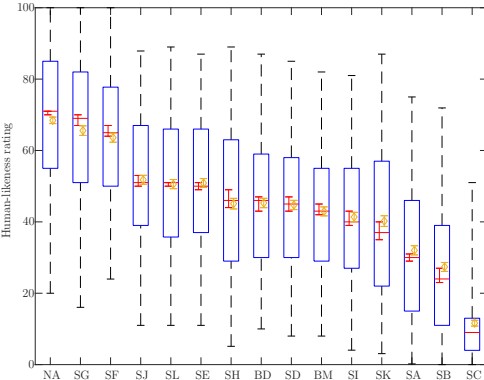

**Figure 2: Box plot visualising the ratings distribution in the human-likeness study. Red bars are the median ratings (each with a 0.05 confidence interval); yellow diamonds are mean ratings (also with a 0.05 confidence interval). Box edges are at 25 and 75 percentiles, while whiskers cover 95% of all ratings for each condition. Conditions are ordered by descending sample median rating.**

latent dimension of the attention-based encoder is 512. The cross-local attention networks use 8 heads, 48 attention channels, the window size is 15 frames (0.5 second), each window looks at the one in front of it, and with a dropout of 0.1. As for self-attention networks are composed of 8 layers, 8 heads, and with a dropout of 0.1. AdamW [23] optimizer (learning rate is $3 \times 10^{-5}$) is used with a batch size of 200 for 1200000 samples. Our models have been trained with $T_d = 1000$ noising steps and a cosine noise schedule. The whole framework can be learned in about 132 hours on one NVIDIA V100 GPU.

### 3.2 Evaluation Setting

The challenge organizers conducted a detailed evaluation comparing all submitted systems [16]. Three proportions were evaluated: human-likeness, appropriateness for agent speech and appropriateness for the interlocutor. We strongly recommend the reference [16] for more details on the evaluation. The following abbreviations are used to denote each model in the evaluation:

- NA: Natural mocap ('NA' for 'natural').
- BM: The official monadic baseline [4], a model based on Tacotron 2 that takes information (WAV audio, TSV transcriptions, and speaker ID) from the main agent as input ('B' for 'baseline', 'M' for 'monadic').
- BD: The official dyadic baseline [4], which also take information from the interlocutor in the conversation into account when generating gesture ('D' for 'dyadic').
- SA–SL: 12 submissions (ours is **SF**) to the final evaluation ('S' for a submission).

### 3.3 Evaluation Analysis

*3.3.1 Human-likeness.* As for human-likeness, participants were asked "Please indicate on a sliding scale how human-like the gesture motion appears". The rating scale from 100 (best) to 0 (worst) is anchored by partitioning the sliders into five equal-length intervals

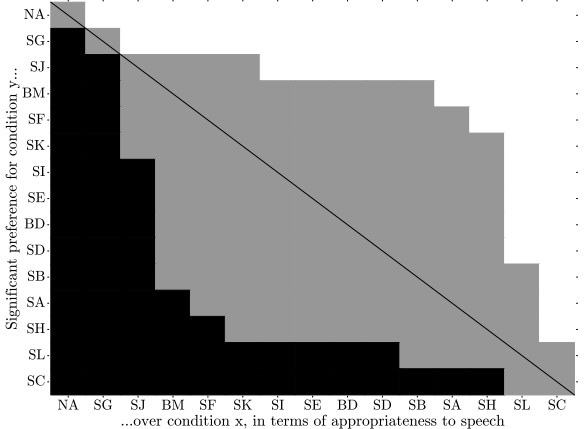

(a) Appropriateness for agent speech

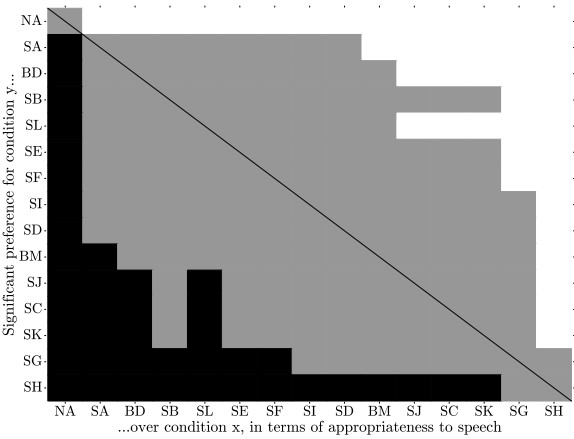

(b) Appropriateness for the interlocutor

**Figure 3: Significant differences between conditions in the two appropriateness studies. White means the condition listed on the $y$-axis achieved a mean appropriateness score significantly above the condition on the $x$-axis, black means the opposite ($y$ scored below $x$), and grey means no statistically significant difference at level $\alpha$ = 0.05 after correction for the false discovery rate.**

labeled "Excellent", "Good", "Fair", "Poor", and "Bad". Bar plots and significance comparisons are shown in Figure 2. The median of our system (SF) was 65 ∈ [64, 67] and the mean was 63.6±1.3. And the human-likeness was not significantly different from the system SG [16]. This result shows that our model can generate very high-quality gestures, but somewhat lower than natural mocap, with a median of 71 ∈ [70, 71] and a mean of 68.4±1.0.

*3.3.2 Appropriateness for agent speech.* In terms of appropriateness for agent speech, participants were asked "Which character's motion matches the speech better, both in terms of rhythm and intonation and in terms of meaning?" Five response options are available, "Left is clearly better", "Left is slightly better", "They are equal", "Right is slightly better", and "Right is clearly better".

**Table 1: Ablation studies results. '+' indicates additional modules and ↔ indicates the length of the modality in the time dimension. Bold indicates the best metric.**

| Name | FGD on feature space ↓ | FGD on raw data space ↓ |
|---|---|---|
| Ours | **14.461** | **531.172** |
| + Seed gesture ↔ $N$ + Speech ↔ $N_{seed}$ (DiffuseStyleGesture [33]) | 19.017 | 767.503 |
| + Seed gesture ↔ ($N + N_{seed}$) | 15.539 | 616.437 |

The mean appropriateness score (MAS) of the submitted system is close to each other, so we report significant differences as shown in Figure 8(a). Our system (SF) with a MAS of 0.20±0.06 and a Pref. matched (identifies how often test-takers preferred matched motion in terms of appropriateness) of 55.8%, which is significantly better than submitted systems SH, SL and BC. However, it has significant deficiencies with natural mocap (NA) with a MAS of 0.81±0.06 and a Pref. matched 73.6% and SG.

*3.3.3 Appropriateness for the interlocutor.* Additionally, an interlocutor who converses with the previous main agent is added to this user interface for scoring. Please ref to [16] for more details. As for appropriateness for the interlocutor, participants were asked "In which of the two videos is the Main Agent's motion better suited for the interaction?". The response options were the same as before, i.e., "Left is clearly better", "Left is slightly better", "They are equal", "Right is slightly better", and "Right is clearly better". We also report significant differences as shown in Figure 8(b). Natural mocap (NA) with a MAS of 0.63±0.08 and a Pref. matched of 69.8% is significantly more appropriate for the interlocutor compared to all other conditions. Our system (SF) with a MAS of 0.04±0.06 and a Pref. matched of 51.5%, which is significantly more appropriate than conditions SG and SH, and not significantly different from other conditions. And our system does not use interlocutor information and (as expected) is not significantly different from chance.

## 3.4 Ablation Studies

Moreover, we conduct ablation studies to address the performance effects of different architectures in our model. We use Fréchet gesture distance (FGD) [37] as the objective evaluation metric, which is currently the closest to human perception among all objective evaluation metrics [17]. The lower FGD, the better. The FGD is computed using the autoencoder provided by the challenge organizers. Our ablation studies, as summarized in Table 1, indicate that when the input of [33] is used (the information of seed gestures and speech is given directly over the full length of a training sample), both metrics perform worse; when additional seed gestures are given over the full length of a training sample on our model, both metrics also become worse. The purpose of using seed gestures [33, 37] is to smooth the transition between generated segments, so they should not contain speech information and should only be considered at the beginning for consistency with the previously generated gestures. We also learn that although the diffusion model has the ability to learn useful information from redundant representations, careful design of the network structure of the denoising module can further improve performance.

## 3.5 Discussion

*3.5.1 Takeaways.* Our co-speech gesture generation model (SF), based on the diffusion model, exhibits comparable levels of human-likeness and appropriateness for the interlocutor when compared to the best performing models (SG, SA). Furthermore, it achieves competitive performance with the leading model (SG) in terms of appropriateness for agent speech. These findings suggest that our proposed model performs at a top-tier level. Our model achieves good results due to the ability of the diffusion model to generate high-quality gestures and the local attention-based structure to generate gestures that correspond to the current short duration of speech. Notably, based on the diffusion model, this can easily generate diverse gestures since the main part of the input is noise and any seed gesture can be set. Moreover, based on the structure of the diffusion model, we add random masks to the denoising module, which enables the interpolation and extrapolation of conditions such as speaker identity (style), and a high degree of control over the style intensity of the generated gestures. However, stylization and diversity are not included as one of the evaluation dimensions in the challenge.

*3.5.2 Limitation.* Our model does not consider the information of the interlocutor, this is also not significantly different from a random selection. Taking into account information about the interlocutor is important in the interaction, and this is a direction for future research. Moreover, pre-processing the data should make the results better. We do not do anything special with motions that do not include movement in the hand and still train with its hand, which can lead to poorer hand results. For the exploration of the dataset and more discussion, please refer to the Appendix.

*3.5.3 More Discussion.* We also tried to add the BEAT [21] dataset (all of them / some of the speakers) to train together with Talking With Hands, but we got worse results, the model didn't converge. We guess the possible reason is that the BEAT dataset is very large, and the diffusion model needs more time to be trained well.

Although we did not consider interlocutors, in terms of appropriateness for the interlocutor, our system (SF) is significantly more appropriate than SG and SH, and not significantly different from other conditions. It is worth noting that SG is the best-performing model on the first two dimensions of the evaluation. We suspect that the reason for this is related to the setting of the evaluation, cause "segments should be more or less complete phrases" in the evaluation. However, the evaluation during silence is equally important, and the model should learn the behavior from the data when not talking, such as idling and other small gestures, and no other unexpected actions. Although we did not consider the information of interlocutors, it is impressive that our model is able to remain idle while the other person is talking (when he/she is not talking).

The diffusion model takes a long time to train and inference. The evaluation was performed using 8-10 seconds of speech, and longer speech evaluation results may be more consistent with human perception. When the number of participants in the speech appropriateness evaluation was 448, there was no difference between our system (SF) and SG; when the number of participants in the evaluation was increased to 600, SG was significantly better than all of the submitted systems, which suggests the differences

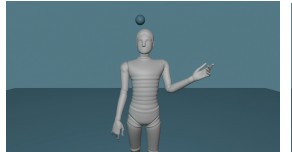 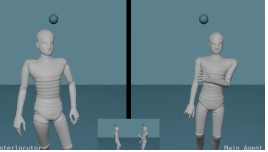

(a) A gesture indicating largeness.    (b) A pointing gesture.

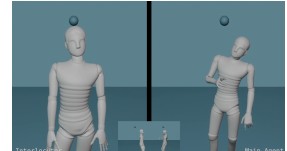

(c) A thinking gesture.

**Figure 4: Case study of generated gestures. The right side of each figure shows the generated gestures.**

between the two systems were relatively small and needed to be statistically significant until a large number of subjects had been recruited and evaluated after FDR correction.

*3.5.4 Case Study.* Our diffusion-based method can extract semantic information and generate human-like gestures. For instance, when the speaker says "large", our system generates a gesture indicating largeness. When the speaker asks "Where do you stay?" our system generates a pointing gesture, mimicking human behavior.

Our diffusion-based models can generate incidental actions for laughter and surprise. For example, when the speaker laughs, the model generates a body shake, mimicking human laughter. When the speaker is thinking, the model generates a corresponding thinking action. This suggests that diffusion-based models can learn semantics and synthesize semantic actions in specific situations.

## 4 CONCLUSION

In this paper, we propose DiffuseStyleGesture+, a diffusion model based method for speech-driven co-gesture generation. Based on the DiffuseStyleGesture framework, we add text modality and then more logically designed the input architecture of the modality, while tuning the representation of gesture and audio according to the challenge dataset to be able to generate high-quality, speech-matched, speaker-specific stylized, and diverse gestures and to be highly controllable on these conditions. The proposed model is in the first tier in human-likeness and appropriateness for the interlocutor, with no significant difference from the best model, and achieves competitive performance with the best model on appropriateness for agent speech, showing the effectiveness of the proposed method. However, compared with the natural mocap, there is still much room for improvement worth further exploration.

## ACKNOWLEDGMENTS

This work is supported by National Natural Science Foundation of China (62076144), Shenzhen Science and Technology Program (WDZC20200818121348001) and Shenzhen Key Laboratory of next generation interactive media innovative technology (ZDSYS2021062-3092001004).

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

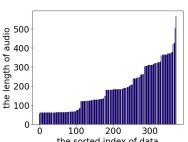 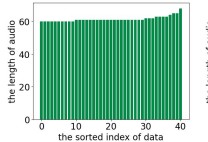 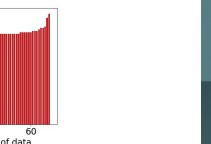

(a) The length of audios in training dataset.

(b) The length of audios in validate dataset.

(c) The length of audios in testing dataset.

**Figure 5: GENEA Challenge 2023 dataset audio length analysis.**

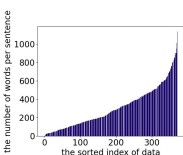 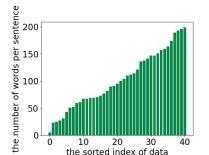 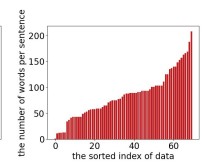

(a) The number of words per sentence in the training dataset.

(b) The number of words per sentence in the validate dataset.

(c) The number of words per sentence in the testing dataset.

**Figure 6: GENEA Challenge 2023 dataset text length (words per sentence) analysis.**

## A    APPENDIX

### A.1    Exploratory Data Analysis

The GENEA Challenge 2023 provided 372 training data, 41 validation data, and 70 test data. The training and validation datasets include text, audio, and BVH motion capture files for both the main agent and the interlocutor. The test data lacks the main agent's BVH motion capture file, which our system aims to predict. Metadata files with speaker identity information are also provided.

*A.1.1    Audio Analysis.* As shown in Figure 5, the duration of the training data varies, ranging from less than 2 minutes to nearly 10 minutes (9 minutes and 27 seconds). The validation and test sets have an average duration of about 1 minute. The total duration of all datasets is approximately 20 hours and 49 minutes.

*A.1.2    Text Analysis.* As shown in Figure 6, the maximum number of tokens in a single piece of training data is 1135. The distribution of data is non-uniform across all types of datasets. Word-frequency statistics were also performed. The three most common words in the dataset are 'like', 'I', and 'Yeah', each used nearly 10,000 times. Laughing is marked with '#', while other emotions such as surprise, silence and other states are not marked.

*A.1.3    Gesture Analysis.* As shown in Figure 7, we identified several issues with the original dataset. Most notably, the upper body of most human figures appears to recede (tilt back), especially in side views. Many speakers exhibit unnecessary foot movement. Some datasets also contain severe bone position errors.

### A.2    Appropriateness Studies

The bar plots for the appropriateness analysis are shown in Figure 8. In terms of appropriateness for agent speech, SG was significantly

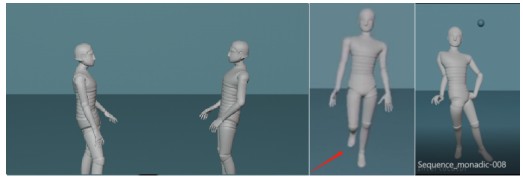

**Figure 7: Some possible problems with the dataset. Better performance may be obtained if the data is preprocessed.**

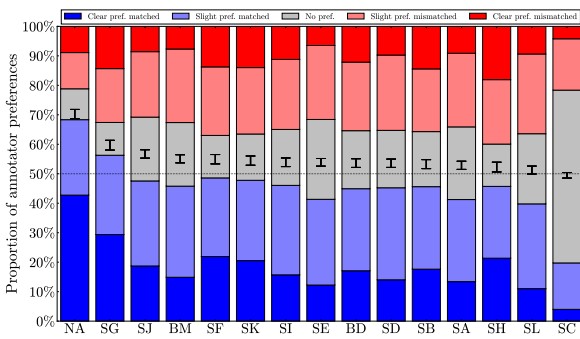

(a) Appropriateness for agent speech

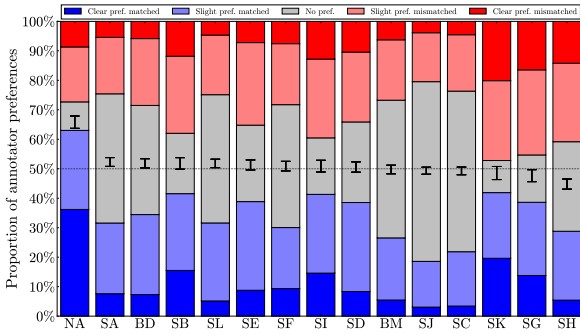

(b) Appropriateness for the interlocutor

**Figure 8: Bar plots visualizing the response distribution in the appropriateness studies. The blue bar (bottom) represents responses where subjects preferred the matched motion, the light grey bar (middle) represents tied ("They are equal") responses, and the red bar (top) represents responses preferring mismatched motion, with the height of each bar being proportional to the fraction of responses in each category. Lighter colors correspond to slight preference, and darker colors to clear preference. On top of each bar is also a confidence interval for the mean appropriateness score, scaled to fit the current axes. The dotted black line indicates chance-level performance. Conditions are ordered by mean appropriateness score.**

higher than our system (SF); in terms of appropriateness for the interlocutor, all systems were not significantly different from random results, or even inferior to random selection. Overall all systems are less appropriate than natural mocap (NA).