# OpenReview forum: "The DiffuseStyleGesture+ entry to the GENEA Challenge 2023"
_ACM.org/ICMI/2023/Workshop/GENEA_Challenge — GENEA Challenge 2023 Mainproceeding_

### Official Review · Reviewer_YLQp · 2023-07-27
**good results by diffusion based method**

**Rating:** 10
**Confidence:** 5

**Review:**

The paper described an updated method of previously proposed diffusion-based gesture generation model. The method improved input network structures, input feature representation, and cross-local attention mechanism of denoising model.

The paper is well organized and written.
The paper properly refers recent works of gesture generation by data-driven approach.
The technical descriptions are well written and the experiments would be reproducible.

The paper updated only minor points but it is valuable to see the results of applying the diffusion based method to this dataset.

**Nominate For A Reproducibility Award:**

diffusion based approach is one of the promising approach of gesture generation. worth a reproduction.

---

### Official Review · Reviewer_byU7 · 2023-08-02
**A good paper describing a well-performing monadic challenge submission**

**Rating:** 8
**Confidence:** 4

**Review:**

Greatest strengths:
* The paper is well written gives and relevant details for replicating and learning from the work, as system-description papers should
* The work is of particular interest because the associated system demonstrated particularly competitive human-likeness in the evaluation
* The appendix providing observations about the dataset, some negative results, etc., adds value
* Releasing code and pre-trained models, as the abstract commits to, promises to be of great use to the community

Main weakness:
* The claims (e.g., abstract line 20 and discussion line 425) that there were no statistically significant differences to top-tier models in terms of appropriateness for the interlocutor are technically correct, but they can easily be read to imply top-tier interlocutor awareness, when in fact the system does not appear to be aware of interlocutor behaviour at all. To instead state that the degree of interlocutor awareness was not statistically different from chance would be equally correct and less likely to suggest an interlocutor-ware system. The authors should consider presenting the results regarding appropriateness for the interlocutor in a more nuanced manner.

-----

Below follow a few detailed comments on the submission, in order of the appearance:

> Line 88: "we find that the diffusion model-based approach for co-speech gesture generation surpasses existing methods in terms of quality, style, diversity, and alignment with speech."

Two comments:
1) The proposed approach is unlikely to have surpassed the motion-graph based approach of GestureMaster in the previous GENEA Challenge, since that achieved better human-likeness ratings than the mocap itself. The statement starting on line 88 should be qualified to be more clear which existing methods are being referred to, e.g., "surpasses other deep generative models of motion in terms of...".
2) "we find that" The results/findings reported in the current paper do not allow concluding that diffusion models are better than other deep generative approaches in terms of style and diversity. It is probably the truth, but is not a finding of the studies in the paper. For example, it is not clear which other models in the challenge evaluation that were diffusion models.

> Line 97: "GENEA 2023 challenge"

The official way to write this is "GENEA Challenge 2023"

> Line 155: "We used 62 joints including the fingers"

How were BVH files without finger mocap handled? (Optionally, if the authors think their handling of missing finger mocap was notable for the results they achieved, they might choose to comment on that somewhere.)

> Line 242: "We train on the official training dataset of Talking With Hands [18] provided by GENEA 2023 [16]"

Two remarks:
1) The GENEA Challenge 2023 dataset is not the official training set of Talking With Hands In fact, this reviewer believes that the Talking With Hands data actually is not partitioned into training and test sets at all. It would be better to write something like "We trained on the official training dataset of the GENEA Challenge 2023 [16], which is based on Talking With Hands [18]."
2) Since the Talking With Hands mocap data is not always perfect, it would be useful to write something like "We trained on all the data in the GENEA Challenge 2023 training dataset", to be unambiguously clear that no data was excluded due to poor mocap, missing fingers, or other reasons. (Conversely, if training data was excluded, the text should specify that, and how the selection was performed.)

> Line 267 (and elsewhere): "NA: Ground truth"

This reviewer recommends against using the term "ground truth" in gesture generation, since (unlike many classical machine learning problems) there is no single, "true" way to move, even to a given speech. "Natural mocap" is one alternative that can be used instead of writing "ground truth".

> Line 328: "3.3.3 Appropriateness for the interlocutor"

The reviewer materials also mention testing whether or not appropriateness is different from chance, by looking at whether or not the MAS confidence intervals overlap with zero. Given that all challenge submissions were close to chance performance in this evaluation, it might be more useful to consider whether or not one's system performed significantly different from chance when discussing the results of this particular user study. It is the understanding of this reviewer that the presented system did not use interlocutor information, and (consistent with expectations) was not significantly different from chance. It would be good to mention this.

> Line 771: "When the number of participants in the speech appropriateness evaluation was 448, there was no difference between our system (SF) and SG"

This wording makes it sound like a difference did not exist until additional subjects were recruited. That is not quite right. The difference was always there between the two systems, and it was always the same size (the systems did not change when the number of test takers was expanded); it is just that the difference was relatively small, and therefore it did not become statistically significant in the evaluation after FDR correction until a large number of subjects were recruited. Tweaking the wording could make the statement in the paper more accurate.

**Nominate For A Reproducibility Award:**

The paper descriptions are clear. If code and pre-trained models are released, then this should be considered a candidate for the reproducibility award.

---

### Official Review · Reviewer_ahVj · 2023-08-02
**Review of the DiffuseStyleGesture+ entry to the GENEA Challenge 2023**

**Rating:** 8
**Confidence:** 3

**Review:**

The modified version of DiffuseStyleGesture framework integrates textual input and additional speech and gesture representations. The evaluation results indicate that this model is able to generate high-quality gesture motions comparing with the ground-truth motions and the other submitted systems.


Comments and questions,

Speech features: Lines 172-173:
Can you explain the benefit of incorporating the supplementary speech features into the network? It would be helpful to compare the Fréchet gesture distance (FGD) results when including and excluding each speech representation.

Text features: Lines 181-184:
There are various actions such as laughter, silence, surprise and so on. Could you explain more about the motivation/benefit of including only laugh information in the text features? How to get laugh information from the text input?


Discussion: Lines 88-94
Both speech and textual input are used in DiffuseStyleGesture. However, there is no mention of the advantages of using the textual modality. It would be interesting to include an ablation test to discuss the impact of textual modality on the results.

Reproducibility: Lines 24-26
The authors will release the code and pretrained models after the acceptance of the paper.


**Nominate For A Reproducibility Award:**

The authors will release the code and pretrained models after the acceptance of the paper.

---

### Decision · Program_Chairs · 2023-08-04

**Decision:**

Accept (Main proceeding)

**Comment:**

All reviewers favoured accepting this paper, and the scores are high (10, 8, 8). The chairs agree to accept this paper to the Main ICMI Proceedings.

Please read the reviews carefully and use the feedback to revise the paper for the camera-ready version, as many of them can make the paper even better. We particularly request that the authors nuance how they present the appropriateness for the interlocutor, to reduce the risk of misinterpretations that the system is interlocutor aware (and hence that its appropriateness differs from random chance).